# Cryo-EM structures of full-length integrin αIIbβ3 in native lipids

Brian D. Adair[1,2], Jian-Ping Xiong[1,2], Mark Yeager [3,4,5] & M. Amin Arnaout [1,2] ✉

Platelet integrin αIIbβ3 is maintained in a bent inactive state (low affinity to physiologic ligand), but can rapidly switch to a ligand-competent (high-affinity) state in response to intracellular signals ("inside-out" activation). Once bound, ligands drive proadhesive "outside-in" signaling. Anti-αIIbβ3 drugs like eptifibatide can engage the inactive integrin directly, inhibiting thrombosis but inadvertently impairing αIIbβ3 hemostatic functions. Bidirectional αIIbβ3 signaling is mediated by reorganization of the associated αIIb and β3 transmembrane α-helices, but the underlying changes remain poorly defined absent the structure of the full-length receptor. We now report the cryo-EM structures of full-length αIIbβ3 in its apo and eptifibatide-bound states in native cell-membrane nanoparticles at near-atomic resolution. The apo form adopts the bent inactive state but with separated transmembrane α-helices, and a fully accessible ligand-binding site that challenges the model that this site is occluded by the plasma membrane. Bound eptifibatide triggers dramatic conformational changes that may account for impaired hemostasis. These results advance our understanding of integrin structure and function and may guide development of safer inhibitors.

Integrins are αβ heterodimeric receptors participating in cell–cell and cell–matrix adhesion in metazoa. Integrins are normally kept on the cell surface in a default inactive state, allowing, for example, platelets to circulate freely in blood vessels without causing potentially fatal blood clots. Physiologic agonists acting through their cognate receptors, promote the binding of intracellular talin and kindlin to the β-subunit cytoplasmic tail, eliciting structural rearrangements in the associated α and β transmembrane (TM) domains that hair-trigger a rapid conformational switch of the large ectodomain to the ligand-competent state ("inside-out" activation). The binding of physiologic ligands initiates "outside-in" signals transduced via the TM domains to the cytoplasmic tails leading to regulated cell adhesion[1]. Excessive integrin activation is associated with multiple pathologies making integrins a prime therapeutic target. However, current integrin inhibitory drugs are partial agonists, inadvertently inducing durable

outside-in signaling, leading to paradoxical adhesion and adverse outcomes in treated patients[2], underscoring the need for a better understanding of the structural basis of bidirectional signaling in full-length integrins.

Many of the details about integrin structure and function came initially from studies of the ectodomains of the β3 integrins αVβ3 and αIIbβ3 in their free and ligand-occupied states. The β3 integrin ectodomain comprises 12 subdomains assembled into a ligand-binding "head" that comprises a seven-bladed β-propeller of the α-subunit and a vWFA-like βA domain of the β3 subunit. The integrin head is followed by upper- (an α-subunit thigh domain, and a hybrid-, PSI- and EGF1 domains of β3) and lower "leg" domains (tandem calf-1 and calf-2 domains of the α-subunit and EGF2-4 and βTD domains of β3). X-ray structures of the unliganded β3 integrin ectodomain revealed an inactive bent structure[3,4] also detected on the cell surface of resting

[1]Leukocyte Biology and Inflammation Laboratory, Structural Biology Program, Division of Nephrology, Department of Medicine, Massachusetts General Hospital, Boston, Massachusetts 02114, USA. [2]Harvard Medical School, Boston, MA 02115, USA. [3]The Phillip and Patricia Frost Institute for Chemistry and Molecular Science, University of Miami, Coral Gables, FL 33146, USA. [4]Department of Chemistry, School of Arts and Sciences, University of Miami, Coral Gables, FL 33146, University of Miami, Miami, FL 33146, USA. [5]Department of Biochemistry and Molecular Biology, Miller School of Medicine, University of Miami, Miami, FL, USA. ✉e-mail: aarnaout1@mgh.harvard.edu

cells[5,6], in which the head and upper leg domains (the headpiece) fold back onto the lower legs at an α- and a β3 genu (knee).

The integrin ligand-binding site comprises an "Arg" pocket at the top of the propeller domain and a *metal-ion-dependent-adhesion-site* (MIDAS), normally occupied by proadhesive $Mg^{2+}$ ion in the βA domain, flanked by two regulatory metal binding sites a *ligand-associated metal binding site* (LIMBS or SyMBS) and an *adjacent to* MIDAS (ADMIDAS), each normally occupied by a $Ca^{2+}$ ion[7,8]. The $Ca^{2+}$ at ADMIDAS links the activation-sensitive N-terminal α1 and C-terminal α7 helices of the βA domain, thus locking βA and hence the integrin in the inactive conformation[3,9]. The ligand-competent state, triggered by inside-out activation, is characterized by a break of the $Ca^{2+}$-mediated link between the α1 and α7 helices, allowing a prototypical Arg-Gly-Asp (RGD)-containing ligand to stably bind the integrin by inserting the Arg side chain into the propeller with the Asp carboxylate making a monodentate contact with the MIDAS $Mg^{2+}$. Bound ligand drives the α1 helix together with the ADMIDAS metal ion inwards towards MIDAS and induce rearrangements in the adjacent loops leading to the downward descent of the α7 helix, which forces a swingout of the underlying hybrid domain. Replacement of the ADMIDAS $Ca^{2+}$ with $Mn^{2+}$ also breaks the α1- α7 link, facilitating the conversion of βA into the ligand-competent state[7,10], but keeps cellular αIIbβ3 in the bent conformation[6].

The X-ray structures of the recombinant β3 ectodomain lead to two models for inside-out integrin activation. The "switchblade" model assumed that the integrin ligand-binding site of the integrin head points down toward the plasma membrane, which blocks its access to physiologic ligand unless the integrin opens first like a switchblade knife[11,12]. The other βTD-centric "deadbolt" model proposed that conformational changes in the TM domains driven by talin/kindlin binding to the β cytoplasmic tail induce movements of the membrane-proximal βTD that break the $Ca^{2+}$-mediated link between the α1 and α7 helices, thus switching the βA domain into the ligand-competent state without the necessity for the switchblade opening[13]. In this model, ligand occupancy induces the hybrid swingout, and genuextension, leading to outside-in signaling. Since the X-ray structures of the β3 ectodomains lacked the transmembrane domains, the essential conduit for bidirectional signaling, determining the actual orientation of the ligand-binding site in a full-length inactive integrin relative to the plasma membrane remains unknown, hampering a better understanding of the structural basis of integrin activation and signaling under physiologic and pathologic states.

Several approaches have previously been taken to obtain the 3D structure of full-length integrin αIIbβ3, the most abundant integrin in platelets. These included rotary shadowing and/or negative stain 2D images of αIIbβ3 in detergent[14–16], cryo-EM of αIIbβ3 in detergent[17], small-angle neutron- or x-ray scattering of αIIbβ3 in detergent[12,18], electron cryotomography of αIIbβ3 in liposomes[19], electron tomography of activated αIIbβ3 in detergent[20], and negative stain EM of αIIbβ3 in lipid bilayer nanodiscs[19,21,22]. Although the globular ectodomain was a consistent feature, other results, including its orientation relative to the plasma membrane, were not concordant (reviewed in ref. [23]). The common use of detergents[24], the dissociation of the αIIbβ3 heterodimer[14], and averaging over heterogeneous conformational states may have accounted in part for these discordant results. NMR structures of the associated αIIb and β3 TM synthetic peptides in detergents or in bicelles[25–27] also revealed differences in the membrane-proximal TM regions, perhaps related to the absence of the ectodomain.

The recent use of detergent-free technologies has emerged as an alternative to detergent-based approaches, allowing extraction and purification of membrane proteins while maintaining a native lipid bilayer environment suitable for EM analysis[28]. In this work, we use the membrane mimetic <u>N</u>ative <u>C</u>ell <u>M</u>embrane <u>N</u>anoparticles <u>P</u>olymer 7b (NCMNP7b, also known as NCMNP7-25)[29], to extract native

αIIbβ3 from platelet membranes and perform cryo-EM on single particles that now recapitulate the native membrane environment, which is far superior to solubilization and purification of native integrins using detergents. We show that native full-length αIIbβ3 adopts an inactive bent conformation but with separated TM α-helices and an extracellular ligand-binding site that is fully accessible to macromolecular ligands. Full-length αIIbβ3 bound to the partial agonist drug eptifibatide undergoes dramatic conformational changes in the integrin that may contribute to the increased bleeding seen in treated patients.

## Results and discussion

The cryo-EM structure of native full-length αIIbβ3 adopts the bent conformation seen previously in the crystal structures of the ectodomain, with all 12 extracellular subdomains of the ectodomain as well as the TM α-helices including the connections with the ectodomain visible in the cryo-EM map (Figs. 1a–c and 2, Supplementary Figs. 1, 2, and 5, and Supplementary Table 1). No density is detected for the αIIb and β3 short cytoplasmic tails (CT). The structure of the bent ectodomain of full-length αIIbβ3 is superimposable onto that of the X-ray structure of recombinant αIIbβ3 ectodomain[4], with an RMSD between the respective 1,592 Cαs of 1.662 Å. The βA domain has clear densities for three metal ion pockets at MIDAS, ADMIDAS, and LIMBS (Fig. 1d and Supplementary Fig. 3a). We assigned the metal ion densities as $Mg^{2+}$ at MIDAS and as $Ca^{2+}$ at ADMIDAS and LIMBS, as previously shown[4]. Densities corresponding to four $Ca^{2+}$ ions at the base of the propeller and at the α-genu are also visible (Fig. 1e). The βA domain assumed the inactive conformation with maintenance $Ca^{2+}$-mediated link between its α1 and α7 helices (Supplementary Fig. 3a), and is superimposable onto the respective domain from the X-ray structure of unliganded αIIbβ3 ectodomain (3fcs.pdb) with RMSD for all respective βA domain Cαs of 0.818 Å. Clear densities are also visible in the cryo-EM map of full-length αIIbβ3 for N-glycans at $N^{15}$, $N^{249}$, $N^{570}$, $N^{680}$, and $N^{931}$ of αIIb and $N^{99}$, $N^{320}$, $N^{371}$, $N^{452}$, $N^{559}$, and $N^{654}$ of β3 (Fig. 1f and Supplementary Table 2). A closeup of the unsharpened map surrounding residues 419–422 of the propeller with the atomic model is shown in Fig. 1g. Densities for the β3 genu ($E^{476}$-$Q^{483}$) and between EGF2-thigh, βTD-hybrid, and EGF3-calf1 domains are shown in Supplementary Figs. 4a–d. Some of these densities were absent or weak in the X-ray crystal structure of the recombinant bent αIIbβ3 ectodomain (3fcs.pdb). The presence of these features suggests that the presence of the TM α-helices plays an important role in improving the overall stability of the bent inactive conformation. An additional globular density, also absent in the X-ray structure of the recombinant αIIbβ3 ectodomain, is wedged between the propeller, calf1, and EGF3 domains (Supplementary Fig. 4d, e), surrounded by a basic amino acid cluster from these three domains, a rare arrangement in 3D protein structures[30]. The basic residues are not conserved in the αV subunit. The proximity of this strong density to the binding site for the activating mAb PT25-2 (Supplementary Fig. 4e) suggests that its displacement may be involved in αIIbβ3 inside-out activation.

Significantly, the cryo-EM map of the full-length integrin displayed clear densities for the αIIb and β3 TM α-helices (Figs. 1 and 2 and Supplementary Fig. 5), but the resolution of the TM domains was insufficient to assign side chains with confidence. To evaluate the two additional densities seen in the TM region, we performed a 3D map classification focusing on the TM helices and the adjacent calf2 and βTD (Supplementary Fig. 5). All four resulting 3D classes contained the same two features indicating that these additional densities do not arise from different orientations of the TM helices. The map resolution in this region is insufficient to determine the chemical composition of these two densities. We have, therefore, tentatively assigned the density close to the outer membrane leaflet and the other that is close to the inner membrane leaflet to cholesterol (Fig. 2), an abundant lipid in platelet membranes[31].

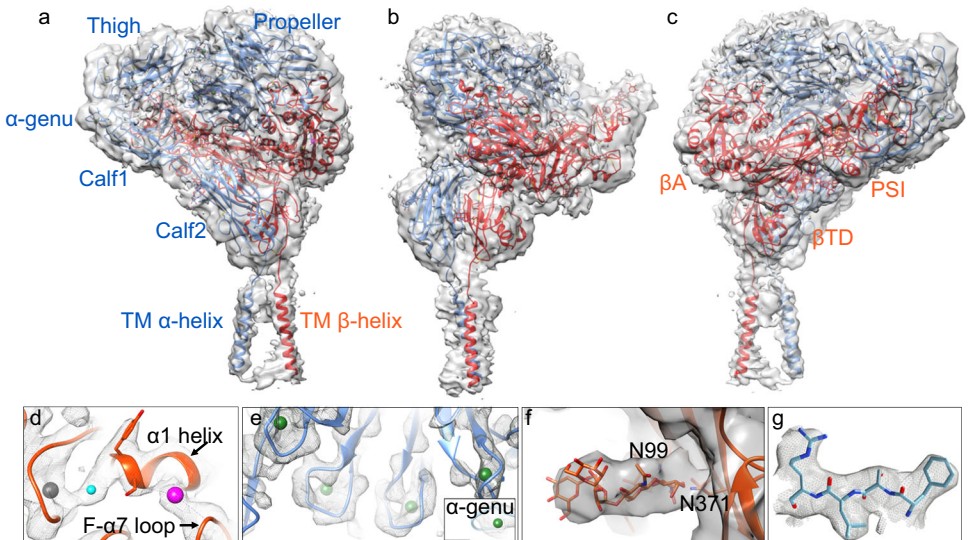

**Fig. 1 | Structure of inactive full-length integrin αIIbβ3. a** The overall unsharpened cryo-EM map at 3.4 Å resolution is shown with ribbon diagrams for the αIIb chain (light blue) and β3 chain (orange-red) here and in subsequent figures. The molecule is oriented with the extracellular side of the membrane facing up. The αIIb and β3 domains and the α-genu are labeled in the respective color, except for the hybrid and EGF1-4 domains of β3. **b, c** The same figure in a rotated by −90° and 180°, respectively. All the 12 subdomains of the ectodomain are resolved, and the two TM α-helices are clearly visualized. **d, e** Closeup of the cryo-EM density of the metal ions (spheres) at LIMBS (gray), MIDAS (cyan), and ADMIDAS (magenta) of the βA domain (**d**), the four metal ions (in green spheres) at the bottom of the propeller (**e**) and the one at the α-genu (**e**, inset). **f** Closeup view of the map for glycans at N⁹⁹ and N³⁷¹ of the hybrid domain. **g** Closeup of residues 419–422 of the propeller domain showing the fit into the EM density.

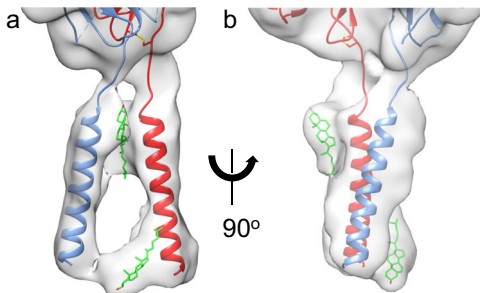

**Fig. 2 | Structure of TM domains with bound lipids of the inactive full-length αIIbβ3. a** Closeup view of the TM region of the cryo-EM map in the same orientation as in Fig. 1a. **b** a 90° rotation of the orientation in **a**. The map has been subjected to a low pass filter at 7.0 Å to correspond to the local resolution of this region. The filtered map shows clear densities for both αIIb (light blue) and β3 (red) TM α-helices. The current resolution does not allow the assignment of side chains for the TM helices with confidence. Alignment with the NMR structure (2K9J.pdb) suggests that the αIIb and β3 TM α-helices end in the new structure at G991 and L718, respectively. The additional densities in the map have been modeled as cholesterol molecules (carbons in green, oxygen in red) at the outer and inner membrane leaflets.

A surprising feature of the full-length structure is the separation of the αIIb TM α-helix from that of β3 (Fig. 2 and Supplementary Fig. 6). There is no density corresponding to the interhelical TM contacts seen in the NMR structure of the αIIb/β3 TM synthetic peptide complex (2K9J.pdb)[25] and shown to keep the native integrin in its inactive state[32] (Fig. 2 and Supplementary Fig. 6). The disordered CT may play a role in forming this new state (combining the bent ectodomain conformation with separation of the TM α-helices). It is known that filamin A, a critical intracellular integrin inactivator[33], acts by engaging and stabilizing both αIIb and β3 CT, thereby restraining αIIbβ3 in the inactive state[34]. We suggest that the new conformation of the full-length αIIbβ3 may represent an intermediate state incurred by the absence of filamin A and before the engagement of β3 CT by talin; the latter interaction

likely further alters the interhelical interface and incurs movements in the βTD, transitioning the ectodomain to the ligand-competent state.

A second feature of the full-length structure is the orientation of the bent ectodomain relative to the plane of the membrane, which fully exposes the ligand-binding site to extracellular macromolecular ligands (Fig. 3 and Supplementary Fig. 7). This orientation challenges a conventional view that inside-out activation of αIIbβ3 requires a switchblade-like opening, and likely explains the ability of cellular bent β2 integrins to bind physiologic ligands[35]. Consistent with mutational studies in cellular αIIbβ3[36,37], the full-length αIIbβ3 structure shows no direct contact of the βTD with the βA domain, proposed by the deadbolt model to form a conformational barrier to inside-out activation of β3 integrins[13], as it does in β2 integrins[38]. The structure, however, shows an EM density connecting the βTD to the hybrid domain (Supplementary Fig. 4c) that is directly linked to the activation-sensitive α7 helix of the βA domain. Computational and mutational studies suggest that eliminating this contact by mutating R633 of the βTD increased the motion of this domain and promoted the binding of αIIbβ3 to fibrinogen to a degree similar to that induced by mAb PT25-2 in wild-type αIIbβ3[39]. Thus, the regulatory role of the βTD in the inside-out activation of αIIbβ3 is perhaps allosteric in nature and reversible in the absence of ligand with the termination of the local TM/βTD movements triggered by agonists.

A third finding revealed in the eptifibatide-bound structure is the large conformational changes in the full-length αIIbβ3 induced by this ligand-mimetic drug. Bound eptifibatide triggered a dramatic 70° swingout of the hybrid/PSI domains (Fig. 4a, b and Supplementary Figs. 8 and 9) previously observed in the X-ray structure of eptifibatide in complex with the truncated αIIbβ3 headpiece bound to a Fab (2VDN.pdb)[40]. A closeup of the unsharpened map surrounding residues 419–422 of the propeller with the atomic model is shown in Fig. 4c. Surprisingly, only the headpiece without the thigh domain was visible in the cryo-EM map of the full-length αIIbβ3-eptifibatide complex (Fig. 4a), despite the lack of evidence for proteolytic degradation (Supplementary Fig. 8f). The absence of additional features corresponding to the leg domains is thus most likely due to drug-induced conformational flexibility. This flexibility is so pronounced that no

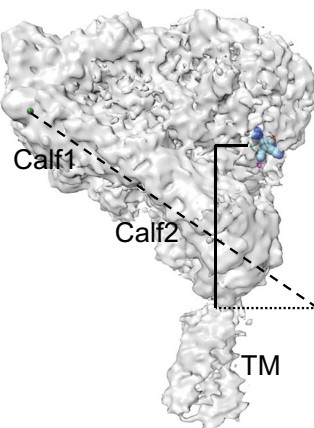

**Fig. 3 | Exposure of the ligand-binding site in the inactive bent conformation of full-length αIIbβ3.** The cryo-EM density map has been fitted with the bound γ-peptide from fibrinogen (as spheres) according to the X-ray structure of the headpiece (2VDR.pdb). The distance measured between the MIDAS metal ion (cyan sphere) and αIIb's Arg[962] at the extracellular leaflet of the membrane (solid vertical line) is 71.5 Å. The calf1-calf2 leg domain axis (dashed line) is tilted from the normal by a 56° angle relative to the plane of the membrane (dotted line). The tilt angle is measured by a chord running from the metal ion at the αIIb genu (green sphere) to Ser[913] at the bottom of the calf2 domain (dotted line).

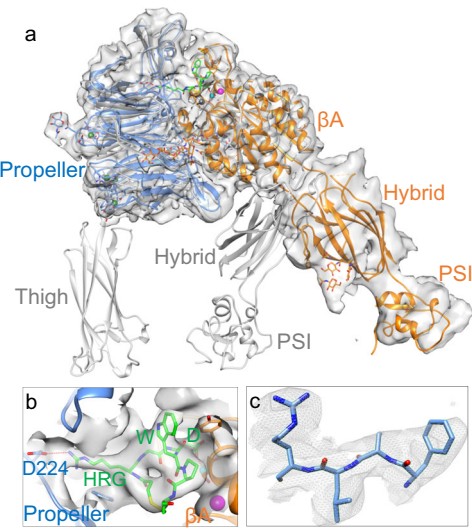

**Fig. 4 | Structure of full-length αIIbβ3 bound to the drug eptifibatide.** Unsharpened cryo-EM map of full-length αIIbβ3 (αIIb in light blue and β3 in orange). The map at the overall resolution of 3.1 Å shows only a density for the integrin αIIb propeller, βA-, hybrid- and PSI domains of the headpiece, but densities are not visible for the thigh, leg, and TM domains, indicating that these domains display conformational variability. In **a**, the αIIb propeller is superposed onto that of the full-length inactive cryo-EM structure (gray) to show the large swingout of the hybrid domain of the eptifibatide-bound integrin. Eptifibatide is shown in green. The metal ions (spheres) at LIMBS (gray), MIDAS (cyan), and ADMIDAS (magenta) of the βA domain are shown. **b** Closeup view of the EM density for bound eptifibatide and the surrounding 6 Å density zone in a different viewing angle from that in **a**. Eptifibatide's homoarginine (HRG) makes a salt bridge (red dotted line) with the propeller's D224, and its Asp (D) monodentately coordinates (red dotted line) the Mg²⁺ ion (cyan sphere) at MIDAS. The eptifibatide structure fits that reported in the crystal structure of the eptifibatide/αIIbβ3 headpiece complex (2VDN.pdb), except that the tryptophan (W) side chain was moved to better fit into the observed EM density. The ADMIDAS Ca²⁺ ion is shown (magenta sphere). **c** Closeup of residues 419–422 of the propeller domain showing the fit into the EM density.

reference-free 2D class average possessed features attributable to the leg domains. The present structure also indicates that the αIIbβ3 TM and CT domains do not limit the global conformational changes induced by eptifibatide and presumably by other partial agonist drugs. The magnitude of these conformational changes makes it less likely that a complete switch of the integrin back to the ligand-competent or inactive state is achievable when the drug disengages from the integrin.

The prothrombotic intense activation of αIIbβ3 under pathophysiologic conditions such as ischemic heart disease can be fatal if untreated. Parenteral use of the three FDA-approved αIIbβ3 inhibitors abciximab (ReoPro®), eptifibatide (Integrilin®), and tirofiban (Aggrastat®), has been shown to be effective in reducing the incidence of myocardial infarction and death in patients with acute coronary syndromes and in reducing the risk of restenosis after percutaneous coronary intervention[41]. However, the direct agonist effect of these parenteral drugs on the integrin is associated with severe thrombocytopenia[42], and oral αIIbβ3 inhibitors are abandoned because of paradoxical coronary thrombosis, thought to be caused by the dissociation of the drug from the integrin, leaving the receptor in a durable ligand-competent state that engages circulating fibrinogen causing platelet aggregation. Elucidating the global structural changes that occur when eptifibatide binds full-length αIIbβ3 could facilitate developing drugs that stabilize the integrin in its inactive state, thus preventing serious on-target adverse outcomes.

## Methods

### Purification of αIIbβ3 in lipid nanoparticles

Outdated platelets were obtained from the Brigham and Women's Hospital blood bank. Platelets were centrifuged at 1500 × *g* for 20 min at room temperature and resuspended in TBS (50 mM tris pH 7.4, 150 mM NaCl) supplemented with 5 mM sodium citrate. Platelets were then centrifuged and resuspended in TBS (without citrate), and the spin was repeated. Platelets were resuspended in HBSS (50 mM Hepes, pH 7.4, 150 mM NaCl), spun again, and resuspended in HBSS supplemented with 1 mM CaCl₂ and 1 mM PMSF. Cells were lysed with three passes on a French press at 800 psi. Cell debris was removed by centrifugation at 1500 × *g* for 20 min, and membranes were collected by centrifugation at 200,000 × *g* for 20 min. Membranes were resuspended in 8 ml of 50 mM HEPES pH 7.4, 500 mM NaCl, 5% glycerol, 1 mM CaCl₂ supplemented with protease inhibitors (Calbiochem inhibitor cocktail III without EDTA, 1 mM PMSF, 10 μM each of Ilomastat, GI 254023X, and TAPI-2). Membranes were solubilized by adding 2% w/w of the NCMNP7b polymer and incubated overnight at 4° C with gentle rocking. Insoluble material was removed by two 20-min centrifugations at 1500×*g* and 200,000×*g*. The extract was incubated with 0.5 ml ConA-sepharose that was previously washed in 50 mM HEPES pH 7.4, 500 mM NaCl, 2 mM CaCl₂, 1% NCMNP7b, for 1 h at 4° C with gentle rocking. The resin was centrifuged at 1500 × *g* for 5 min, resuspended in the ConA washing buffer above, and the washing was repeated. Protein was eluted by adding 0.5 ml of 50 mM Hepes, 150 mM NaCl, 1 mM CaCl₂, and 1 M n-methyl maltopyranoside and incubated for 1 h at 4° C with gentle rocking. Protein was dialyzed against two changes of TBS with 1 mM CaCl₂ using cellulose acetate 100,000 MW cutoff membranes at 4° C. Protease inhibitors were added following dialysis, and protein was stored at 4° C. The αIIb and β3 subunits of the purified full-length αIIbβ3 in complex with eptifibatide in NCMNP7b were intact, as demonstrated by SDS-PAGE analysis (Supplementary Fig. 8f). Protein concentration was determined by absorption spectroscopy at 280 nm using a calculated extinction coefficient of 207,000 M⁻¹cm⁻¹ derived from the number of Trp (24) and Tyr (54) residues in the protein and individual extinction coefficients of 5700 M⁻¹cm⁻¹ and 1300 M⁻¹cm⁻¹, respectively. Eptifibatide samples proceeded as above, except that the dialysis buffer consisted of TBS with 1 mM CaCl₂ and

1 mM MgCl$_2$. Eptifibatide was added following dialysis at 12-fold molar excess 48 h before EM sample preparation.

## Cryo-EM sample preparation and data collection

Electron microscopy was performed at the Molecular Electron Microscopy Core at the University of Virginia. Quantifoil 1.2/1.3 grids were glow discharged in a Pelco EasiGlow at 20 mAmp for 45 s in the presence of amyl amine. A 3.5 μL aliquot of platelet integrin in NCMNP7b at 3.4 μM (0.66 mg/ml) was applied to the carbon side of each grid in a Vitrobot Mark IV held at 4 °C and 100% relative humidity. Grids were blotted in the Vitrobot with Whatman #1 filter paper, using a blot force of 7 and blot time of 30 s, and vitrified in liquid ethane cooled by liquid nitrogen. Cryo-EM data were collected on a Titan Krios electron microscope operating at 300 kV equipped with a Gatan K3 summit detector and a GIF Quantum energy filter. Image movies were collected with EPU software (Thermo Fisher) using a slit width of 10 eV on the energy filter and a magnification of 81,000 for a pixel size of 1.08 Å. Image stacks contained 40 frames with an electron dose of 1.25 e$^{-1}$/Å$^2$ per frame. Stacks were motion corrected during collection using Cryosparc Live.

## Cryo-EM data processing

All image processing was performed in CryoSparc 4.2.1[43]. The outline for processing of the native αIIbβ3 is shown in Supplementary Figs. 2. Processing commenced on an initial data collection of 4287 micrographs. Following the CTF assignment, 862,992 particles were selected from templates generated from particles selected by a blob picker and subjected to class averaging. 104,332 particles with class averages that resembled integrin projections were combined and used for ab initio model generation and refinement, which resulted in a map of the bent integrin with weak density attributable to the transmembrane domains. A larger set of particles (348,546 including the originally selected 104,332 particles) selected from the class averages were subjected to heterogeneous 3D refinement into four classes. One of the classes containing 61,409 particles was selected for further processing, generating a map at 3.7 Å (Supplementary Fig. 2).

Additional 3202 micrographs from the original grid and 4455 micrographs from a second grid prepared at the same time from the same sample were collected, and 1,139,722 and 2,475,122 particles were selected, respectively, using templates generated as above. Following class averaging, 283,247 particles from the new data were combined with the 61,409 particles to generate a set of 311,656 particles, and used for heterogeneous 3D refinement with three copies of the 3.7 Å integrin map as starting models. One of the three classes containing 117,126 particles (yellow-colored map in Supplementary Fig. 2) clearly displayed the transmembrane region. These particles were refined with non-uniform 3D refinement, which generated a map with a resolution of 3.4 Å. Observing that the automated masking routines applied with each refinement iteration generated masks that failed to enclose the transmembrane region, a static mask was generated, which enclosed the ectodomain and provided space through the center of the weak lipid nanoparticle density. Using this mask generated an equivalent overall map resolution at 3.4 Å but with clear densities for the αIIb and β3 TM α-helices (Fig. 1a–c).

Processing on eptifibatide-bound αIIbβ3 was performed on a single stack of 9805 micrographs (Supplementary Fig. 9). Following the CTF assignment, 2,645,177 particles were selected from templates generated from particles selected by a blob picker and subjected to class averaging. Of these, six classes containing 434,172 particles were combined and used for ab initio model generation. A heterogeneous refinement against two of the ab initio models generated two maps, each fitting an integrin headpiece without the thigh or the subsequent domains. One of the maps (colored yellow in Supplementary Fig. 9), with 268,647 particles, demonstrated less noise and superior overall resolution and was used for subsequent non-uniform refinement, which resulted in the presented 3.1 Å resolution map.

## Model building

To prepare an initial model of the full-length αIIbβ3, the available crystal structure of the unliganded ectodomain (3fcs.pdb)[4] was fit as a rigid body into the EM density map using PHENIX 1.20.1[44]. The model was further refined by iterative manual building/adjustment in Coot[45] and real-space refinement in PHENIX and REFMAC 5.8.0267[46]. Sharpening the map with an applied isotropic B factor did not improve the quality of the fit. The fit of the individual αIIb and β3 TM domains was done in Chimera 1.16[47] using the NMR structure (2K9J)[25]. The final unliganded structure fitted well to the cryo-EM density with a correlation coefficient (CC mask) of 0.85. For the eptifibatide-bound full-length αIIbβ3, the crystal structure of the headpiece (2VDN.pdb) was used for initial docking in PHENIX. As with the unliganded structure, B factor sharpening did not improve the fit. Eptifibatide and glycans were modeled in Coot, and the resulting model was refined through real-space refinement with secondary structure restrains from the 2VDN.pdb structure using PHENIX with a final CC mask is 0.78.

## Reporting summary

Further information on research design is available in the Nature Portfolio Reporting Summary linked to this article.

## Data availability

The EM maps and atomic coordinates for the full-length unliganded αIIbβ3 and the eptifibatide-bound αIIbβ3 have been deposited in the EMDB (www.ebi.ac.uk/pdbe/emdb/) and Protein Data Bank (www.rcsb.org) under the accession codes EMD-40989 (unliganded structure), 8T2V (unliganded structure), EMD-40988 (eptifibatide/αIIbβ3 structure), and 8T2U (eptifibatide/αIIbβ3 structure). Publicly available PDB entries used in this study are available under the accession codes 3FCS, 2VDR, 2VDN, 3EIO, C9JEU5, and 2K9J.

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

## Acknowledgements

We thank Kelly Dryden and Michael Purdy of the University of Virginia cryo-EM facility for expert technical assistance. Transmission electron micrographs were recorded at the University of Virginia Molecular Electron Microscopy Core facility (RRID: SCR_019031), which is partly supported by the School of Medicine. In addition, the Titan Krios (SIG S10-RR025067), Falcon II/3EC direct detector (SIG S10-OD018149), and K3/GIF (U24-GM116790) were purchased in part or in full with the designated grants. We thank Dr. Youzhong Guo, Virginia Commonwealth University, Richmond, VA, for providing the NCMN P7b polymer. We thank Dr. Michael Hanson, University of Miami, Miami, FL, and Dr. Johannes van Agthoven, Massachusetts General Hospital, Boston, MA,

for helpful discussions. This work is supported by National Institutes of Health grants HL141366, DK088327 (M.A.A.).The cryo-EM structure of native full-length αIIbβ3 has been presented at the Gordon conference on "Fibronectin, Integrins, and related molecules" on 7 Feb 2023.

## Author contributions

M.A.A. conceived the project and designed the experiments. M.Y. suggested using styrene-maleic acid copolymer lipid particles (SMALPs) for solubilization. B.D.A. conducted protein purification and performed cryo-EM data collection and image processing. Molecular modeling was performed by J-P.X., and M.A.A.; M.A.A. supervised the project and wrote the manuscript with input from all co-authors.

## Competing interests

M.A.A. is a co-founder of a 2021 startup aimed at generating and testing pure integrin antagonists. The other authors declare no competing interests.
