## [Peer Review File · Nature Communications]

Cryo-EM Structures of Full-length Integrin α IIb β 3 in Native LipidsREVIEWER COMMENTS

Reviewer #1 (Remarks to the Author):

The manuscript by Adair, Xiong, Yeager, and Arnaout adds an important contribution to our understanding of integrin structure and function

Notably, the full-length integrin structure determination has been sought for decades, and it is incredibly exciting to have this full-length integrin structure that includes the entire transmembrane domain and the linker region to the ectodomain

This is a fantastic contribution the field has been waiting for decades

The eptifibatide-based allostery is exciting, and it would be great to add or cite supporting biochemical data

Some rewriting would be helpful to display the importance of their significant findings better

1. Please address the resolution:

a. For example, some statements are over beyond what the current resolution can provide with certainty, while key findings (for example, that the ligand binding site is accessible or a potentially unprecedented pre-primed state) are not fully explored

b. For example, please remove “high-resolution” from the title

Perhaps something like

“CryoEM structures of full-length platelet integrin suggest a pre-primed state with an exposed ligand binding site”

c. Likewise, please remove “well-defined” in the legend of Figure S4 when describing the map, and please add the color coding to Figure legends

d. The maps do not seem to match the 3.2 Å, and 3.6 Å reported resolution

Please comment on PMIDs 27552081 and 26424086 for pitfalls of GSFSC that might apply here

e. Please comment on one of the triaged 2D classes resemblances to the eptifibatide

b. Please improve the presentation:

a. The manuscript appears to be put together in a rush; perhaps they were up against a competing group

b. Please run a spell and grammar check; for example, “The map at 3.6Å resolution shows only a density

for the integrin headpiece but no densities are visible for the leg or TM domains”

c. Please add a few paragraphs at the start of the introduction to tie the study back to the biological relevance that motivated this structure determination in the first place

d. Please add a final paragraph on the biological implications of health-related significance

c. Please improve the figures:

a. Figure 2 – the disclaimer about the lipid identification is helpful – has the possibility of multiple conformations causing the additional volume been considered?

b. Please show where eptifibatide binds in Figure 4

c. Please label the small molecules, depicted side chains, and anions in Figure S3 and perhaps remove the water molecules

d. Please explain the asterisk in Figure S4 and color coding; please enlarge panels to fill the page and to identify panels outside the image area

e. Please add more panels and information to Figure S6

f. Please label domains in Figure S7 and please, add the information about the rotations to the Figure, and indicate where the 11.4 Å is being measured (from what atom to what atom)

g. Figure S8 could be improved: perhaps view from the top or the top at an angle if the fibrinogen-integrin interaction is not visible when viewed from the top; the geometry of the helices of fibrinogen could be corrected to show as helices; integrin might be better shown as a cartoon instead of the volume

h. Please add a scale bar to the micrographs in both Figures

d. Please improve the understanding:

a. Please quantify this statement “The overall ectodomain arrangement largely agrees”

b. Please consider deleting this statement “assuring the successful reconstitution of the full-length integrin into the native lipid environment.” as the crystal structures are not necessarily the physiological states of the protein

c. Please rewrite this sentence, perhaps splitting it into two sentences and specifying residue ranges to provide more clarity:

“Except for contacts in the region immediately following the ectodomain-TM linker, for which there is visible density, the interhelical TM inner- and outer contacts seen in the NMR structure of the α IIb/ β 3 TM peptide complex, shown to keep the integrin in its inactive state 29 are not visible in our map”

d. Please describe the unprecedented procedure mentioned here “We used a custom membrane mimetic to extract α IIb β 3 from platelet membranes as native cell-membrane nanoparticles” in more detail in the methods as well as in the results section

e. Please provide the full name of “NCMN P7b” and where it was purchased from

f. Please explain the Goldilocks concept to the scientist outside the field

g. Please comment on the effect of using 1 mM manganese on the activation of integrin

Reviewer #2 (Remarks to the Author):

This manuscript reports the full length platelet integrin α IIb β 3 structure using cryoEM technique. There are a significant number of integrin extracellular domain structures including α IIb β 3. There are also low resolution (20A) full length α IIb β 3 structure and high resolution structure of α 5 β 1 with and without fibronectin ligand. However, this is the first structure in membrane media that revealed the density of transmembrane domain with different assembly topology than the previously reported structure of isolated transmembrane domain and also some extracellular subdomains that were missing in previous crystal structures. Overall, the extracellular domain structure and orientation vs membrane are different from previous crystal structures, suggesting new ligand binding mechanism. Moreover, the authors reported the drug bound state of the integrin structure providing some potential for the therapeutic development. I feel the new structural information is quite significant but have a few questions:

1. How certain are the authors in defining the end regions of α IIb and β 3 transmembrane domains with the current resolution? The authors should make the text clear where exactly the transmembrane domains end in their map given that the cytoplasmic regions are not visible. If the resolution does not allow the definition of the end residues of the transmembrane domains, the authors should mention that.
2. Can the authors provide an overlay of the extracellular domain in this structure with previous crystal structure, which may help readers to understand the difference more clearly? This can be provided in a supplementary figure.
3. The new structure revealed that the ligand binding site is fully accessible suggesting that ligand may

induce conformational change of the receptor but a large number of previous studies demonstrated that talin is clearly needed to disrupt the α IIb/ β 3 cytoplasmic tail association and trigger integrin activation/ligand binding. Can the authors discuss this in a bit more detail about their thoughts? Is it possible that talin releases the α IIb/ β 3 association restraint in the cytoplasmic side to facilitate fibrinogen to bind the “accessible” site to allow a global conformational change and produce a fully extended/active state?

We deeply appreciate the very positive feedback we received from both reviewers on the original manuscript. Their valuable comments helped to improve this manuscript further. Following is a point-by-point response to each comment raised (italicized) and the places in the revised version (in blue text) where additions or modifications were made.

Reviewer #1 (Remarks to the Author):

The eptifibatide-based allostery is exciting, and it would be great to add or cite supporting biochemical data.

We have added a Coomassie stain of SDS-PAGE showing intact α IIb and β 3 subunits of the eptifibatide-bound full-length receptor, indicating that the absence of the additional domains in the map is secondary to drug-induced large conformational flexibility and not to proteolytic cleavage of the subunits (page 20, lines 1-3 and new Supplementary Figure 8f).

Some rewriting would be helpful to display the importance of their significant findings better.

1. Please address the resolution:

(a). For example, some statements are over beyond what the current resolution can provide with certainty, while key findings (for example, that the ligand binding site is accessible or a potentially unprecedented pre-primed state) are not fully explored.”

We have analyzed additional particles of the full-length integrin, which provided improved densities for both TM α -helices (page 7, para.2, lines 1-2; page 21, para.2, Figs 1a-c and Fig. 2).

We have addressed in two paragraphs the potential significance of the separated TM domains and the accessible ligand binding site in the bent state (page 8, para. 1 and para 2, respectively).

b. For example, please remove “high-resolution” from the title. Perhaps something like “CryoEM structures of full-length platelet integrin suggests a pre-primed state with an exposed ligand binding site.”

We have removed “high resolution” from the original title. The suggestion that the new structure may represent a pre-primed state was proposed in the original abstract.

c. Likewise, please remove “well-defined” in the legend of Figure S4 when describing the map, and please add the color coding to Figure legends.”

The “well-defined” phrase has been removed from the legend of Supplementary Figure 4, and color coding has been added to the Figure legends.

d. The maps do not seem to match the 3.2 Å, and 3.6 Å reported resolution Please comment on PMIDs 27552081 and 26424086 for pitfalls of GSFSC that might apply here.”.

We are confident that the quality of our maps does not suffer from the pitfalls in the two quoted references (use of too many particles or the averaging of large numbers of dissimilar molecules). In support, new Fig.1g and Fig. 4c, show the fit of residues 419-422 of the propeller into the respective EM density. Reference to these figures has now been made in the revised text (page 7, para.1, lines 1-2, and page 9, para. 1, lines 5-7)

We agree that the isosurface used to display the two maps do not match the respective global resolutions as determined by the GSFSC. This is now shown by comparing the different isosurfaces in Supplementary Figure 1e (upper vs. the added lower panels). We elected to display the map in Fig. 1a-c with an isosurface level that clearly displays the TM regions, albeit at a lower

resolution (see Supplementary Figure 1e). Similarly, for Figure 4, the isosurface was chosen to display the bound drug and the PSI domain.

e. *“Please comment on one of the triaged 2D classes resemblances to the eptifibatide.”*

We have analyzed additional particles for the eptifibatide/integrin complex, which now clearly show eptifibatide in the ligand binding pocket (new Fig. 4b, and page 9, para.1, line 4).

b. Please improve the presentation:

a. *“The manuscript appears to be put together in a rush; perhaps they were up against a competing group.”*

The reviewer is correct. We were rushing to submit it before the senior author gave an invited talk at the Gordon conference on Feb 7th, 2023, presenting the EM structure of native full-length α IIb β 3 (added to the acknowledgment section, page 11, lines 1-3).

b. *“Please run a spell and grammar check; for example, “The map at 3.6Å resolution shows only a density for the integrin headpiece but no contain densities are visible for the leg or TM domains”*

Spell and grammar checks have been run.

c. *“Please add a few paragraphs at the start of the introduction to tie the study back to the biological relevance that motivated this structure determination in the first place.”*

Three new paragraphs have been added at the beginning of the revised manuscript to tie the study back to the biological significance as the reviewer suggested. These additions dictated minor adjustments to the abstract and text to avoid duplications.

d. *Please add a final paragraph on the biological implications of health-related significance.*

As the reviewer suggested, a final paragraph on the biological implications of health-related significance has been added at the end of the revised manuscript.

c. *Please improve the figures:*

a. *“Figure 2 – the disclaimer about the lipid identification is helpful – has the possibility of multiple conformations causing the additional volume been considered?”*

Yes. This possibility has been considered and addressed (page 7, para. 1, lines 2-7, and revised Supplementary Figure 5).

b. *“Please show where eptifibatide binds in Figure 4.”*

This is now clearly shown in revised Fig. 4b.

c. *“Please label the small molecules, depicted side chains, and anions in Figure S3 and perhaps remove the water molecules.”*

The ligand binding site of the final 3.4 Å map of the full-length integrin does not contain a small molecule. Key activation-sensitive-side chains are labeled. The water molecules have been removed (Supplementary Figure 3a).

d. *“Please explain the asterisk in Figure S4 and color coding; please enlarge panels to fill the page and to identify panels outside the image area.”*

The asterisk in Supplementary Figure 4d and e indicate the new density. Color coding of the respective α IIb and β 3 domains is now added to Supplementary Figure 4 legend, Supplementary Figure 4 is enlarged, and the subdomains are labeled in the respective panels.

e. *“Please add more panels and information to Figure S6.”*

The original Supplementary Figure 6 has been relabeled as Supplementary Figure 8 to better follow the text. A larger number of 2D class averages of the eptifibatide-bound integrin has now been added (Supplementary Figure 8b, page 21, para. 2); these resemble one another due to the limited number of orientations, as shown in the added panel (Supplementary Figure 8d). For comparison, we have added an equivalent panel to Supplementary Figure 1 (panel d), showing the more complete distribution of orientations for the unliganded integrin.

f. *“Please label domains in Figure S7 and please, add the information about the rotations to the Figure, and indicate where the 11.4 Å is being measured (from what atom to what atom).”*

The original Supplementary Figure 7 has been relabeled Supplementary Figure 6 to follow the text better. The relative rotations are now shown, and how the RMSD was measured is also indicated (Supplementary Figure 6 legend).

g. *“Figure S8 could be improved: perhaps view from the top or the top at an angle if the fibrinogen-integrin interaction is not visible when viewed from the top; the geometry of the helices of fibrinogen could be corrected to show as helices; integrin might be better shown as a cartoon instead of the volume.”*

The original Supplementary Figure 8 has been relabeled Supplementary Figure 7 to follow the text better. A closeup of the ligand binding site occupied by the fibrinogen ligand is now added as an *inset* to improve visibility.

h. *“Please add a scale bar to the micrographs in both Figures.”*

Scales have been added to Supplementary Figures 1a and 8a.

d. *Please improve the understanding:*

a. *Please quantify this statement “The overall ectodomain arrangement largely agrees.”*

The overall similarity (RMSD) is now specified (page 6, para. 1, lines 5-8).

b. *“Please consider deleting this statement “assuring the successful reconstitution of the full-length integrin into the native lipid environment.” As the crystal structures are not necessarily the physiological states of the protein.”*

This statement has been deleted.

c. *“Please rewrite this sentence, perhaps splitting it into two sentences and specifying residue ranges to provide more clarity: “Except for contacts in the region immediately following the ectodomain-TM linker, for which there is visible density, the interhelical TM inner- and outer contacts seen in the NMR structure of the α IIb/ β 3 TM peptide complex, shown to keep the integrin in its inactive state 29 are not visible in our map.”*

This sentence has been rewritten (page 8, para. 1, lines 1-5).

d. *“Please describe the unprecedented procedure mentioned here “We used a custom membrane mimetic to extract α IIb β 3 from platelet membranes as native cell-membrane nanoparticles” in more detail in the methods as well as in the results section.”*

We have added a new reference (#29) describing the compound used in solubilization. The procedure for nanoparticles is not otherwise different from methods for solubilizing membranes with detergents.

e. *“Please provide the full name of “NCMN P7b” and where it was purchased from.”* The full name is Native Cell Membrane Nanoparticles Polymer 7b (NCMNP7b) (page 5, last line). It is a gift from Dr. Youzhong Guo (acknowledged).

f. *“Please explain the Goldilocks concept to the scientist outside the field.”* The additional three paragraphs (requested by reviewer 1) and 150-word limit of the abstract made expanding this concept superfluous and has been removed from the abstract and text.

g. *“Please comment on the effect of using 1 mM manganese on the activation of integrin”* A phrase has been added on page 4, para. 1, last line.

Reviewer #2 (Remarks to the Author):

1. *“How certain are the authors in defining the end regions of α IIb and β 3 transmembrane domains with the current resolution? The authors should make the text clear where exactly the transmembrane domains end in their map, given that the cytoplasmic regions are not visible. If the resolution does not allow the definition of the end residues of the transmembrane domains, the authors should mention that.”*

This has now been made clear in Fig.2 legend.

2. *“Can the authors provide an overlay of the extracellular domain in this structure with previous crystal structure, which may help readers to understand the difference more clearly? This can be provided in a supplementary figure.”*

A major difference between the cryo-EM map of the native full-length α IIb β 3 and the crystal structure of its recombinant ectodomain is the absence of the critical β genu from the latter structure. We have added a new panel (b) to Supplementary Figure 3 and on page 6, para.1, line 9, demonstrating this important difference and the precise location of the β genu within EGF2, thus correcting relevant literature.

3. *“The new structure revealed that the ligand binding site is fully accessible, suggesting that ligand may induce conformational change of the receptor but a large number of previous studies demonstrated that talin is clearly needed to disrupt the α IIb/ β 3 cytoplasmic tail association and trigger integrin activation/ligand binding. Can the authors discuss this in a bit more detail about their thoughts? Is it possible that talin releases the α IIb/ β 3 association restraint in the cytoplasmic side to facilitate fibrinogen to bind the “accessible” site to allow a global conformational change and produce a fully extended/active state?”*

This is exactly what the β TD-centric deadbolt model of inside-out activation proposed: the ligand binding site in the integrin head is accessible to the ligand in the bent state, but requires prior activation. This activation is triggered by talin-induced movements in the TM helices and the proximal β TD domain, that either directly (as in β 2 integrins) or allosterically (as in β 3 integrins) breaks the link between β A’s activation-sensitive α 1 and α 7 helices, thus priming the β A domain. This is now detailed in the third introductory paragraph requested by reviewer 1 (page 4, para. 1).

REVIEWERS' COMMENTS

Reviewer #1 (Remarks to the Author):

The manuscript by Adair, Xiong, Yeager, and Arnaout adds an important contribution to our understanding of integrin structure and function

Notably, the full-length integrin structure determination has been sought for decades, and it is incredibly exciting to have this full-length integrin structure that includes the entire transmembrane domain and the linker region to the ectodomain

The revision makes up for the hasty presentation of their original submission which greatly improved the manuscript

While the new gel of the eptibatide-bound integrin (new Figure S8d) does not show that eptibatide is bound, the manuscript has improved significantly and this is a minor point now

#1 – resolution:

The resolution concerns have been addressed and I applaud the authors for their thoroughness in analyzing additional particles that ended up being a worthwhile endeavor

Their additional Figures S1e, 1g, 4b, and 4c add a lot of value to addressing this point

#2 – presentation:

Spelling – please check “genuextension”

“pro adhesive” might be better than “proadhesive”

please chose closeup or close-up but not both, capitalize or (better) not Propeller but don't use both

perhaps “non priming” instead of “nonpriming”

Figure legends 1 and 4, please consider replacing “orange red” with “dark orange” or “orange-red”

Grammar –

please add an “s” to remain in “the impact of the TM domains on structural stability remain unknown”

please choose “another partial agonist” or “other partial agonists” instead of “other partial agonist”

please replace “at” with “on” in “Color encoding for particle numbers is shown at the right”

punctuation -

please remove the comma in “Densities for the β 3 genu (E476-Q483), and”

please add a comma before “as” in “to form a conformational barrier to inside-out activation of β 3 integrins 13 as is the case in β 2 integrins”

please remove the comma in “the regulatory role of the β TD in the inside-out activation of α IIb β 3 is perhaps allosteric in nature,”

please add a space after “4” in “4°C”

please remove the comma in “The model was further refined by iterative manual building/adjustment in Coot 45,”

please remove the comma in “a color-coded local resolution map of the integrin (in a similar orientation to that in Figure 4), is generated using the 0.143 cutoff criterion.”

vocabulary –

please replace “full” with “complete” or “entire” in “The full map” (legend S5)

#3 – Figures:

Please use the space available in the supplement and please help the reader with more labels

For example (with additionally labeling beta1, beta7, etc. and the other ions), see attached

Please label the atoms shown in sticks in S4a as done in the other S4 panels and increase the asterisk size

Please add more depth to the zoomed-in box that is now added to Figure S7 and label molecules and

domains and indicate the location of the membrane as two (interrupted by integrin) lines and use cartoons for secondary structure in fibrinogen

Please state what residues are included in the RMSD calculations (e.g. Figure legend S6) and indicate the membrane as two (interrupted by integrin) lines and label the atoms shown in sticks (Figure S6)

#4 – clarity:

It would still be helpful to state what residues cannot be modeled due to weak density

What is the structure and composition of NCMNP7b and how does one generate it if it is not commercially available? Without this information, one will not be able to reproduce the data presented in this manuscript; please provide the details

Miscellaneous (and very minor)

I am a little confused that the PDB entries (x, y, and z) are unknown, as this should have already been deposited when obtaining the structure report

Reviewer #2 (Remarks to the Author):

I am satisfied with the authors' responses.

We again deeply appreciate the very positive feedback we received from reviewers on the revised manuscript. Following is a point-by-point response to the comment raised (italicized) and the places in the revised text or supplementary Fig. legends (in blue text) where changes were made.

Reviewer #1 (Remarks to the Author):

#1 – resolution:

“The resolution concerns have been addressed and I applaud the authors.”

Thanks.

#2 – Presentation

Spelling:

Please check “genuextension.”

Checked.

“pro adhesive” might be better than “proadhesive”

Will keep proadhesive as it is used as such in the literature.

“please chose closeup or close-up but not both, capitalize or (better) not Propeller but don’t use both”

closeup and lower case propeller have been used throughout.

“perhaps “non priming” instead of “nonpriming”

“non priming” is now removed from abstract.

Figure legends 1 and 4, please consider replacing “orange red” with “dark orange” or “orange-red”

“orange-red” is now used.

Grammar:

please add an “s” to remain in “the impact of the TM domains on structural stability remain unknown”

Done.

please choose “another partial agonist” or “other partial agonists” instead of “other partial agonist”

“other partial agonist drugs” is correctly used.

please replace “at” with “on” in “Color encoding for particle numbers is shown at the right”

Done.

punctuation -

please remove the comma in “Densities for the $\beta 3$ genu (E476-Q483), and”

comma removed.

please add a comma before “as” in “to form a conformational barrier to inside-out activation of $\beta 3$ integrins 13 as is the case in $\beta 2$ integrins”

comma added.

please remove the comma in “the regulatory role of the β TD in the inside-out activation of α Ib β 3 is perhaps allosteric in nature,”
comma removed.

please add a space after “4” in “4°C”
Space added.

please remove the comma in “The model was further refined by iterative manual building/adjustment in Coot 45,”
comma removed.

please remove the comma in “a color-coded local resolution map of the integrin (in a similar orientation to that in Figure 4), is generated using the 0.143 cutoff criterion.”
comma removed.

vocabulary –

please replace “full” with “complete” or “entire” in “The full map” (legend S5)
Full was replaced with “entire”.

#3 – Figures:

Please use the space available in the supplement and please help the reader with more labels For example (with additionally labeling beta1, beta7, etc. and the other ions), see attached
We have now labeled α 1 and α 7 helices in Supplementary Fig. 3 and indicated the source of the crystal structure in the Fig.3 legend. The metal ions at the bottom of the propeller and in the β A domain are indicated in Supplementary Fig. 3 legend; labeling these in the figure will compromise clarity.

Please label the atoms shown in sticks in S4a as done in the other S4 panels and increase the asterisk size
Done.

Please add more depth to the zoomed-in box that is now added to Figure S7 and label molecules and domains and indicate the location of the membrane as two (interrupted by integrin) lines and use cartoons for secondary structure in fibrinogen
We added more depth to the zoomed-in box, which corresponds precisely to the box in the full-length structure, where the TM domains are shown (Supplementary Fig. 7 legend). We, therefore, do not see a need to use cartoons to indicate the location of the membrane.

Please state what residues are included in the RMSD calculations (e.g., Figure legend S6) and indicate the membrane as two (interrupted by integrin) lines and label the atoms shown in sticks (Figure S6)
The residues used in RMSD calculations are now listed in Supplementary Fig. 6 legend. We labeled a critical disulfide shown in sticks but removed glycans so as not to reduce clarity. The β 3-subunit is now in orange.

#4 – clarity:

It would still be helpful to state what residues cannot be modeled due to weak density

A phrase has been added as follows: but the resolution of the TM domains was insufficient to assign side chains with confidence (page 5, para.2).

What is the structure and composition of NCMNP7b and how does one generate it if it is not commercially available? Without this information, one will not be able to reproduce the data presented in this manuscript; please provide the details

The structure of NCMNP7b (same as NCMNP7-25) has been published recently (ref. 29). (see page 4, para 2).

Miscellaneous (and very minor)

I am a little confused that the PDB entries (x, y, and z) are unknown, as this should have already been deposited when obtaining the structure report.

We have now deposited the maps and coordinates.